# Assessing Trade-Offs and Optimal Ranges of Density for Life Expectancy and 12 Causes of Mortality in Metro Vancouver, Canada, 1990–2016

**DOI:** 10.3390/ijerph19052900

**Published:** 2022-03-02

**Authors:** Jessica Yu, Paul Gustafson, Martino Tran, Michael Brauer

**Affiliations:** 1School of Population and Public Health, University of British Columbia, 2206 E Mall, Vancouver, BC V6T 1Z3, Canada; michael.brauer@ubc.ca; 2Department of Statistics, University of British Columbia, 3182 Earth Sciences Building, 2207 Main Mall, Vancouver, BC V6T 1Z4, Canada; gustaf@stat.ubc.ca; 3School of Community and Regional Planning, University of British Columbia, 433-6333 Memorial Road, Vancouver, BC V6T 1Z2, Canada; martino.tran@ubc.ca

**Keywords:** density, mortality, urban planning, life expectancy, cause-specific mortality, urban health, growth management, population growth, difference-in-differences, cubic splines

## Abstract

Background: Understanding and managing the impacts of population growth and densification are important steps for sustainable development. This study sought to evaluate the health trade-offs associated with increasing densification and to identify the optimal balance of neighbourhood densification for health. Methods: We linked population density with a 27-year mortality dataset in Metro Vancouver that includes census-tract levels of life expectancy (LE), cause-specific mortalities, and area-level deprivation. We applied two methods: (1) difference-in-differences (DID) models to study the impacts of densification changes from the early 1990s on changes in mortality over a 27-year period; and (2) smoothed cubic splines to identify thresholds of densification at which mortality rates accelerated. Results: At densities above ~9400 persons per km^2^, LE began to decrease more rapidly. By cause, densification was linked to decreased mortality for major causes of mortality in the region, such as cardiovascular diseases, neoplasms, and diabetes. Greater inequality with increasing density was observed for causes such as human immunodeficiency virus and acquired immunodeficiency syndrome (HIV/AIDS), sexually transmitted infections, and self-harm and interpersonal violence. Conclusions: Areas with higher population densities generally have lower rates of mortality from the major causes, but these environments are also associated with higher relative inequality from largely preventable causes of death.

## 1. Introduction

Nearly 70% of the world’s population will be living in urban centers by 2050 [1]. Understanding and managing the impacts of population growth and densification are, therefore, important steps to implementing the 2030 Agenda for Sustainable Development. Growth management plans (GMPs) or urban containment policies, such as the ‘greenbelt’ in Ontario, Canada, and urban growth boundaries in Oregon, United States, have managed residential development outside or inside pre-determined urban–rural boundaries often to support sustainability and climate change agendas. From an energy efficiency perspective, GMPs can help reduce car dependency, support development of public transportation, preserve ecological features, and reduce air pollution [2]. GMPs have some potential downsides and constraints, such as potential gentrification, public opposition (e.g., NIMBY—‘not in my backyard’), and land economics that support sprawl [3]. Even so, many cities around the world have some form of urban growth control. Theories have emerged that suggest there are net benefits and costs for population health from densification. For example, the density–health relationship may be described by an inverted ‘U-shaped’ curve, whereby positive health indicators can broadly increase with population density due to increase in proximity and access of services, until a threshold where densification shifts to overcrowding and services and infrastructure are constrained beyond their capacities [4]. The Lancet series on urban design, transport, and health also highlights the importance of understanding the risks and benefits of densification; achieving optimal levels of residential density is one of the eight integrated regional and local interventions that may help improve urban design and transport mode choices [5]. Without incorporating these risks and benefits in planning and land-use guidelines, planners may direct growth in areas without factoring in potential adverse impacts on population health and equity.

Population density is a form of discrete planning density [6], where the numerator is a discrete item (e.g., number of people) and the denominator is a spatial area (e.g., square kilometer, km^2^) [7]. The concept of density should be differentiated from crowding, congestion, or sprawl [8]. The former can be used as a planning measure for urban growth, while the latter terms imply conditions that negatively affect urban health [4,7]. Density measures are often obtained from census data and used in planning services and infrastructure. To assess for viability of service provision, population density is considered a more reliable measure than residential density (dwellings per square kilometer) because occupancy rates vary by housing types (e.g., high rise buildings compared to single-family homes) [9]. Therefore, net residential density may not correlate with net population density. Proponents of New Urbanism, or dense, mixed-use development, argue that increasing population and residential density are requirements for a vibrant city [8,10]. In this study, the concept of ‘density’ will be operationalised as population density, which is often correlated with other built environmental features, such as the density, proximity, and availability of essential services, employment, and educational opportunities [11,12,13,14,15,16,17,18,19,20].

In population health, there is mixed evidence on the relationship between population density and different health indicators [21]. Previous studies have shown that increasing density can lead to positive mediating effects, such as increasing active transport [12], leisure time physical activity [22], and increased opportunities for social connections, health services, and specialised care [6,23]. However, densification can also lead to negative mediating effects, such as increased noise and traffic-related air pollution, increased access to fast food restaurants [24], increased smoking and alcohol use [21], and decreased access to nature or green spaces [25]. In addition, increased density has been directly associated with schizophrenia prevalence and poor mental health [26,27]. There is, however, less understanding on how these intermediate risk factors interact in the overall density–health relationship. From a land use perspective, this gap in knowledge leads to a policy and planning relevant question, given that there are health trade-offs with increasing densification, can density serve as a policy-relevant measure to find the optimal balance for health?

To address this question, we linked population density with a 27-year dataset in Metro Vancouver that includes census-tract levels of life expectancy (LE), cause-specific mortalities, and area-level deprivation. We applied difference-in-differences (DID) models with the objective to study the impacts of densification changes from the early 1990s on changes in mortality over a 27-year period. Furthermore, we sought to describe the shape of the relationship between density and multiple mortality outcomes to identify the thresholds of densification whereby net health costs outweigh the net benefits.

## 2. Materials and Methods

### 2.1. Overview

We integrated census tract-level socioeconomic status indicators, population density, and life expectancy at birth in 368 census tracts (CTs) in Metro Vancouver, Canada (1990–2016). Using almost three decades of repeated cross-sectional data, we analysed the change in density over the change in LE using DID models to determine what relation, if any, density had with LE and selected cause-specific mortality outcomes. A DID model is a quasi-experimental approach that compares the outcomes of groups exposed to different policies (e.g., growth management plans) and environmental factors (e.g., population densification) at different periods. Furthermore, we fit cubic smoothing splines and their first order derivatives to examine the shapes and optimal ranges of the relationship between population density, life expectancy, and 12 causes of mortality. We present analyses of density-LE stratified by socioeconomic status (SES).

### 2.2. Study Design—Growth Management Plans in the 1990s

The Metro Vancouver region has undergone rapid growth in the past three decades, increasing from a 1990 population of approximately 1.56 million to a 2016 population of approximately 2.46 million [28]. Over 80% of this expansion was outside the urban core. Government authorities in the region have sought different plans and targets to manage growth and improve environmental sustainability [2]. The “growth management” or “urban containment” plans were drafted in the early 1990s to focus growth within the core and inner suburban areas, with directions on urban structure, protected areas, housing, and population and employment distribution. These drafts led to the adoption of the Liveable Region Strategic Plan (LRSP) in 1996, which included a series of policy statements intended to concentrate 70% of the population growth within metropolitan areas by 2021, reduce car dependency and increase transportation choice in the region, and preserve green space. The adoption of the 1996 LRSP set up a benchmark year of this quasi-natural experimental design to assess how neighbourhoods that densified the most in Metropolitan Vancouver compared to neighbourhoods that densified less. We derived the baseline levels from the pre-densification period for each CT, which we define as the years prior to the adoption of the LRSP (from 1990–1995). See Figure 1 for a summary of the study design.

### 2.3. Data Sources and Preparation

Population density was derived from six cycles from the Canadian census [29] and imported into the Population Data BC virtual research environment. Census-tract level life expectancy at birth and cause-specific mortality data were collected from Yu et al. [30], which were derived from small area, mixed effects Bayesian models [31]. The Canadian Material and Social Deprivation Index (MSDI) was based on the Canadian census and was collected from the Canadian Urban Environmental Health Research Consortium (CANUE) [32]. Six different indicators were chosen based on Peter Townsend’s idea of ‘observable and demonstrable disadvantage’ to reflect material deprivation, or the lack of everyday goods and commodities, and social deprivation, the fragility of an individual’s social network. The material deprivation variables include persons without a high-school diploma, ratio employment/population, and average personal income. The social deprivation variables include persons living alone, persons separated, divorced or widowed, and single-parent families. The Canadian census data from earlier years was crosswalked to the 2016 shapefile using Allen and Taylor’s bridge files [33]. All intercensal years were linearly interpolated. We estimated population density by summing the population over the land area in square kilometer. Density and LE estimates were pooled for 1990–2016 in Metro Vancouver and then stratified by deprivation quintile groups and year. Outlier CTs with respect to density that had more than 1.5 interquartile range above the third quartile and below the first quartile were trimmed and excluded from further analysis.

### 2.4. Statistical Analysis

All eligible CTs in Metro Vancouver census metropolitan area region (*n* = 368) were considered for analysis. For the DID models only, treated and control designations were based on the relative change in density from 1990 to 2016. Treated groups were defined as being in the 90th percentile (P90) or greater for density increases. CTs that served as controls were in the 10th percentile (P10) or lower for density increases over this same period. In stratified analyses, we subset CTs in the lowest and highest MSDI quintile groups and re-ran the DID models.

To assess the change in density over time using 1996 as the benchmark year, we used a difference-in-differences (DID) design to model the cross-sectional measures. The DID design assumes confounders varying across groups (all unmeasured covariates that differ systematically between the two groups, such as health behaviours) are time invariant and time-varying confounders (all unmeasured covariates that change between the two time periods, such as age structure of the population) are group invariant. A two group, two period DID model was modeled for each SES class (all/low/high) resulting in a total of three models. See Appendix B for more details on model specifications.

Time series analyses were used to assess for graphical evidence of the common trend assumption (see Appendix A). For interpretation, the difference-in-differences term were categorised in quintiles for changes that resulted in decreases in mortality estimates. For increases in mortality outcomes, the estimates were categorised into tertiles given the fewer outcomes that resulted in this direction.

In addition, to assess the shapes of the relationships between density and the different health outcomes, smoothed cubic splines were used. We calculated the first order derivative of the spline function to estimate where the curvature or slope of the density-mortality relationship changes sign. We use the density point at the lowest mortality rate and the inflection point to identify the optimal range of density. Therefore, the optimal range in this study is where life expectancies are maximised, mortality rates are minimised, and the curvature of the spline changes sign (the inflection point).

## 3. Results

### 3.1. Baseline Statistics

Prior to the adoption of the LRSP in 1996, in the entire region, the median population density in Metro Vancouver was approximately 2400 people per km^2^, the median LE was 79.9 years, and the median material and social deprivation index (MSDI) score was 1.0 (see Table 1). In the first model with all CTs eligible, 37 CTs were included in the treated group and 37 in the control group (see Appendix A for geographic assignment of treatment and control groups). The median baseline population density was approximately 5 times higher for P10 (1915 persons/km^2^) compared to P90 (387), meaning the treatment group contained areas with lower density in 1990–1995, but which increased the most in density by 2016. All other measures (i.e., median life LE, MSDI) were approximately similar at the baseline level between the treatment and control groups. In the second model of low SES (high MSDI scores) CTs only, 19 were included in the treated group and 20 were included in the control group. At baseline, the median population density was approximately around 5.8 times lower (515/2974), median life expectancy was 0.7 years (79/78) higher, and material and social deprivation scores were half (331/763) for those CTs that increased density the most (P90). In the third model of high SES (low MSDI scores) CTs only, 17 were included in the treated group and 16 were included in the control group. For P90 compared to P10, the median population density was on third (5802/1482) lower, the median life expectancy was 2.6 years (78.4/75.7) lower, and median material and social deprivation scores were approximately similar (1.0). Notably, the treatment group of the high-SES CTs started with the lowest LE (2.7 years less than the control group) and the treatment group of the low SES CTs started with a higher LE (0.7 years difference compared to the control group).

### 3.2. Difference-in-Differences (DID) Analyses

In the analysis of changes in density on changes in LE, increasing density led to slightly decreased LE for all CTs and more so for low SES CTs. For low SES CTs, the LE decreases (−0.85 years, 95% CI: −2.6, 0.89) may be attributed to large increases in mortality from cardiovascular diseases (CVD) (3.67 × 10^−4^ per 100,000 people, 95% CI: 4.13 × 10^−5^, 6.94 × 10^−4^), self-harm and interpersonal violence (4.14 × 10^−5^, 4.14 × 10^−5^, 1.57 × 10^−5^), and unintentional injuries (1.2 × 10^−4^, 4.97 × 10^−5^, 1.90 × 10^−4^). In contrast, density increases appeared to be associated with increased LE for high SES CTs (3.34 years, 2.17, 4.51), potentially driven by larger decreases in mortality from CVD (−9.93 × 10^−4^ per 100,000 people, −1.27 × 10^−3^, −7.21 × 10^−5^) and neoplasms (−1.57 × 10^−3^, −3.33 × 10^−3^, 1.19 × 10^−4^). That is, among the highest SES CTs, LE increased for the CTs that densified the most compared to CTs that densified the least. The opposite can be said for low SES CTs—LE decreased for the CTs that densified the most compared to the CTs that densified the least.

In terms of other cause-specific mortality results, mortality from maternal and neonatal diseases and transport injuries increased with increasing density for all CTs, regardless of SES status. In contrast, mortality from chronic respiratory diseases, neoplasms, respiratory infections and tuberculosis, and substance use disorders decreased for all CTs and SES statuses. Overall, higher relative magnitudes of density-driven decreases in mortality rates of almost all causes were observed in high SES CTs compared to low SES CTs, except for HIV/AIDS and STIs and transport injuries.

See Figure 2 for summary of the DID results and Appendix A for complete model statistics.

### 3.3. Shape of the Association for Life Expectancy

To assess the shape of the association between population density and mortality outcomes, cubic smoothing splines were used to model LE and 12 cause-specific mortality functions across all CTs and years. For life expectancy at birth for both males and females, we found a non-linear, declining relationship (see Figure 3a). That is, LE decreased with increasing densification, and the rate of loss in LE accelerated after a certain inflection point (densification threshold). The inflection point, derived from the first order derivative function (see Figure 3b), for the density-LE function was identified at 9400 people per km^2^. The highest LE was identified at the minimum densification thresholds (<10 people per km^2^). Therefore, the optimal density level for LE was identified as minimum density to 9400 people per km^2^. Phrased in another way, we observed higher average LEs and negligible changes from the current minimum thresholds up to 9400 per km^2^. Thereafter, the decreases in LE begin accelerating. Generally, the density-LE function shows that this trade-off is very minimal (<2 years) with increasing densification to the current maximum limits (>30,000 people per km^2^). Most of the data points of LE below 75 years were found in the lowest SES CTs.

Among the lowest SES group, there was an overall declining ‘S-shaped’ relationship between density and LE, where the sign changes twice on the function (see Figure 4a). Increasing density was associated with decreased LE until a density of approximately 8000 people per km^2^, thereafter, LE increased slightly with increasing density through to the inflection point (14,600 people per km^2^). Generally, most of the data points lie before the inflection point. The optimal range of the overall function for low SES CTs was identified as minimum—14,600 people per km^2^.

Among the highest SES group in Vancouver, there was an overall rising ‘S-shaped’ relationship (see Figure 4b). Higher density was associated with high LE until approximately 8000–10,000 people per km^2^, and, thereafter, begins to decrease LE through to the inflection point, which was identified at 15,800 people per km^2^. The optimal range of density of the entire function for high SES CTs was identified at 15,800–31,500 people per km^2^ although it should be noted that there are much fewer data points in this range to support this.

The full results of the smoothed cubic splines and first order derivative functions can be found in Appendix A and Appendix A.

### 3.4. Shape of the Association for Cause-Specific Mortality Rates

For the density and cause-specific functions, most of the data points lie below 15,000 people per km^2^. Figure 5 highlights a select number of splines with the dashed lines representing the ‘optimal ranges’ that were identified. Among all the functions, there were non-linear and declining ‘S-shaped’ relationships for density and mortality rates of cardiovascular diseases (CVD) (optimal ranges: 21,000–34,600 people per km^2^), neoplasms (15,000–34,600), respiratory infections (14,300–34,600), and diabetes and chronic kidney diseases (17,600–25,900). There was also a negative linear relationship between density and transport injuries (17,600–25,900). For these causes, most of the mortality rates were lowest at the highest density ranges.

In terms of other causes, we observed non-linear and rising ‘S-shaped’ relationships for HIV/AIDS and other STIs (412–13,800), maternal and neonatal diseases (min–14,900), self-harm and interpersonal violence (min–12,606), and unintentional injuries (min–13,024). If we focus on where most of the data points lie, we observed mostly flat relationships for chronic respiratory diseases (min–16,800), neurological diseases (9100–18,900), and substance use disorders (min–34,550). For these causes, mortality rates were lowest at the lower or medium density ranges. Phrased in another way, the rate of acceleration of mortality rates begins to increase at lower inflection points identified (e.g., 13,800–18,900). See Appendix B for a detailed table of optimal ranges and remaining cubic spline figures.

To assess for potential bias from observing the same patterns for each year in the pooled data, we applied cubic spline models to life expectancy and a select number of causes for the year 2016. Similar inflection points and optimal ranges were identified for LE and most causes, except for CVDs and neoplasms. The spline for CVD displayed a slight ‘U-shaped’ curve and the optimal range was in the lower density ranges (1000–15,800), while the spline for neoplasms showed a mostly flat relationship and the optimal range was in the middle (10,400–20,100). See Appendix A for results.

## 4. Discussion

### 4.1. DID Model Results by SES

To our knowledge, this is the first study that attempts to draw out the effects of densification on multiple aggregate mortality outcomes by SES (deprivation) and over time using a quasi-experimental design. Our study found that density increased LE for high SES CTs and decreased LE for low SES CTs; the difference can be attributed to the SES differences in mortality from CVD, and to a lesser extent from self-harm, interpersonal violence, and unintentional injuries. We saw that low SES CTs did not benefit from the overall decrease in mortalities from these causes through densification, which was especially unexpected for CVD (~15%) given we observed the largest decrease in the proportion of overall mortalities from this cause during this period [30]. Over time, density decreased mortality from these causes in high SES CTs but increased mortality in low SES CTs. In fact, density decreased mortality rates for almost all causes in higher magnitudes in high SES CTs compared to low SES CTs, except for HIV/AIDS and STIs and transport injuries; the latter cause group increased for all CTs regardless of SES status, which is unsurprising given that there are increased opportunities to interact with vehicles with more people. These findings are interesting given that our previous study [30] showed that median mortality rate and absolute inequality has increased for neurological disorders and diabetes mellitus and kidney diseases in the region but were shown to have density-related decreases in this study. In fact, population density and SES are not the only drivers of widening relative spatial inequalities in the region given that HIV/AIDS and STIs and transport injuries were shown to have some of the highest relative inequalities in the region (17.4 and 5.6, respectively) and we observed higher density-driven mortality decreases from these causes across low SES CTs in this study. These differences may be attributed to the inequitable types of services, employment, and access to natural space that is available with increasing density by SES. For example, the CVD mortality outcomes observed in high SES CTs may be due to well-reported linkages to health-promoting mediators that are available in these CTs, such as increased opportunities for leisure and work-related physical activity [22,34], transit access [35], and nutritional food options [36,37]. In contrast, high dense and low SES CTs may also be linked to higher densities of fast-food outlets [38,39], injury related to violence [40], and crowding [41]. Other spatially varying factors, such as air pollution, which is typically correlated with density may also be relevant [42]. From a planning perspective, these results suggest that a basic level of access to amenities and services needs to be in place before any marginal benefits can be realized with higher density. Additional research is needed to explain the factors underlying the differential mortality impacts observed by SES over time and the mechanisms by which density may enhance SES-driven health inequalities.

### 4.2. Optimal Ranges of Density for Multiple Mortality Outcomes

In our cross-sectional analyses, we found a non-linear relationship between density and mortality, whereby the adverse impacts of increased density would begin to outweigh health benefits at a certain threshold. The inflection point we identified in our study was at approximately 9400 population per km^2^ for life expectancy using data on all CTs. This inflection point is within the range of what was advocated by Patrick Abercrombie in the 1940s; he suggested for neighbourhood units in Greater London of 10,000 people, or a net density of 3000 to 10,000 people per km^2^, with the intention that these neighbour units would feel like villages or small towns where the community spirit is still lively and perceptible [43]. More recently, Sarkar et al. 2017 study in Hong Kong found an inverted ‘U-shaped’ relationship with residential units and adiposity outcomes, whereby density increased adiposity outcomes until a certain threshold (1800 residential units per km^2^ or ~4900 persons/km^2^) and, thereafter, decreased and provided health benefits [44]. In contrast, Yin et al. 2020’s study in China found a ‘U-shaped’ relationship between local population density and waist-hip ratio (WHR), whereby densification lowered WHR until around 15,000 people per km^2^ and a stark reversal occurred [45]. Unlike the two other studies, we did not find a ‘U shape’ or an inverted ‘U-shape’ in the density-cardiovascular mortality relationship using 27 years of data; instead, we found that density was associated with decreased mortality sharply at lower density ranges and then flattened out at higher densities. This observation may be due to the distribution of population densities and, in particular, the small number of CTs with more than 15,000 people per square km^2^. Another possibility is that the relationship may have changed over time as we also observed a slight ‘U-shaped’ relationship using data from 2016 only. To our knowledge, this is the first study in a major metropolitan city in North America to analyse the shape of the relationship with mortality outcomes, and not restricted to cardiovascular morbidity and mediator outcomes, with further differentiation of other causes of death.

Although we observed an inverse relationship between density and life expectancy at birth, there were overall negligible absolute differences (<2 years) in the range of the spline. The limited published literature on density and mortality shows mixed results; much of the literature does not comprehensively assess by cause and SES. A US-based study derived a compactness index at the county level and found that life expectancy was significantly higher in compact counties compared to counties with more sprawl [46]. Two ecologic studies in the Netherlands and one study in Japan reported positive associations between density and all-cause mortality, but they were in rural or middle-sized urban centers and did not assess the shape of the relationship [21,47,48]. By cause, we found higher thresholds of optimal levels (up to 34,600 people per km^2^) for the three major contributors of mortality in the study region: cardiovascular disease, neoplasms, and diabetes and chronic kidney diseases. In contrast, a population-level, longitudinal study in Scania, Sweden found a dose–response association between population density and lung cancer mortality and ischemic heart disease mortality among participants aged 55+ years and older in both urban and rural areas [49]. In our study, we did not disaggregate by age and found that mortality from cardiovascular diseases and neoplasms were overall lower at higher density ranges for all ages in primarily urban areas. In contrast, the causes that contributed to some of the highest relative inequality in the region [30], including HIV/AIDS and STIs, maternal and neonatal disorders (MNNDs), and self-harm and interpersonal violence, were optimal at lower population density ranges (less than 15,000 people per km^2^). The result for MNNDs may not be intuitive given that maternal health coverage was previously shown to increase with density [23]. A previous study found that rising maternal mortality observed temporally in Canada is likely due to improvements in the ascertainment of maternal deaths with the introduction of the ICD-10 coding system [50]. Further investigation is needed to clarify these findings. Our study also observed a mostly flat relationship between population density and chronic respiratory diseases with an optimal range at 14,075–26,208 people per km^2^. Previous studies have indicated that mortality from chronic respiratory diseases is related with higher population density and smoking status but varies by age [51]; higher rates of chronic obstructive pulmonary deaths are more often found among younger populations in urban areas and senior populations in rural areas. Future studies should examine these functions with age-specific rates to further unravel these relationships.

### 4.3. High-Density and Mortality from Respiratory Infections

Moreover, our findings show that the optimal range of population density for respiratory infection mortality is within the higher thresholds (up to 34,600 people per km^2^) in Metro Vancouver. Our DID analysis also found a significant inverse relationship for densification and mortality rates from respiratory infections in high SES CTs. With recent interest in population density as it relates to the spread of respiratory infections, there is novel and mixed evidence on the effects of density on COVID-19 mortality rates [52]. Recent studies in the US found that larger metropolitan areas have higher mortality rates from COVID-19 [53]. That is, connectivity of transit, social, and economic relationship between counties were more important than population density. Other US-based studies in urban areas found that more dense urban areas can lead to early breakouts, but that both the infection and mortality rates were unrelated with urban population density [54,55]. In contrast, studies in India, Algeria, and Turkey found that population density was associated with higher rates of COVID-19 mortality [56,57,58]. Future studies should assess how neighbourhood connectivity and additional correlates may explain these findings and continue to update and monitor the data to include more recent respiratory infection-related deaths.

### 4.4. Limitations

Population density is only one type of discrete, two-dimensional areal measure for planners to use [6]. It includes only daytime population of residential neighbourhoods, thereby excluding daily changes in density levels based on employment and educational centers. Moreover, density changes in this study did not differentiate between land use changes. That is, did neighbourhoods increase population through the addition of the same dwelling units and/or from changes in land permits, such as from single-family zoning to multi-family zoning? Future studies should consider more nuanced understanding of density, including incorporating measures, such as measured or perceived crowding, ‘buffer zones’ around the census output area, land use changes, or floor area space [59,60]. Future studies can also consider net density instead of gross density by including areas that are only residential or mixed-use, thereby excluding commercial areas and parks. Although, the measurement of net density varies across different settings, which makes it difficult for universal application [7]. Arguably, gross density can still be considered a useful summary measure as it is often correlated with several well-known built environmental factors, such as service availability, mixed land use, and lack of natural space [7,61]. Additional research on population density and the inter-dependencies with other spatial attributes, such as densities of various housing types, employment, services, nighttime population, or occupancy, would provide a more nuanced understanding of the overall density–health relationship [6].

As with most spatial analyses, the findings of this study are subject to the modifiable area unit problem (MAUP). That is, the reporting of density exposures or mortality outcomes is subject to the boundaries of the CTs. Since this study used data at the CT level, it is subject to both scale and zoning effects [6]. With scale effects, larger areas will likely have different results since larger geographic areas usually average out the differences. With zoning effects, many of the boundaries of the CTs can be considered arbitrary, and, therefore, will lead to different results in the way they are zoned or merged in this study. As with most spatial analyses, it is difficult to account for potential neighbourhood self-selection.

Population mobility during the densification period can impact CT cumulative exposure and, therefore, lead to potential misclassification, especially of people’s earlier experiences. This bias is especially relevant for the younger populations more than older populations who are more settled. For example, younger migrants have been shown to move longer distances; the migration out of the CT has an effect of decreasing the proportion of healthier populations in the area. However, older migrants tend to move shorter distances to similar environments where there are good medical services. In a previous migration analysis of this mortality dataset [30], we linked 10 years of residential history to the mortality data and assessed how reassigning the CT of exposure based on duration of residency compared to last recorded CT affected the life expectancy measures. The analysis did not show a substantial change in the LE estimates. Nevertheless, mobility of international migrants, who tend to be healthier, and populations moving between CTs of different SES were not accounted for and should be considered in future studies. Unfortunately, annual residential data beyond this temporal scope were not available for this study. Future collection and ongoing surveillance of self-reported residential histories especially during earlier years can potentially help address this limitation.

The decision to use smoothed cubic splines is only the first step to identify a potential optimal range. In the case study of Metro Vancouver, most of the data points were below 15,000 people per km^2^. Hence, the confidence bands are wider at higher densities. Readers should interpret these results carefully. Moreover, for the DID models, there were baseline LE differences for the high SES CTs of around 2.7 years lower in the treatment group compared to the control group, meaning the treatment group started at a lower life expectancy before densification. A large difference in baseline LE may affect the DID results if we expect that improvements in life expectancy from densification is not linear. That is, public and environmental health interventions may be more likely to improve life expectancy from 70 to 75 years than from 85 to 90 years, as an example. For latter age groups, we would expect greater contribution and advancements from healthcare and science to help people live longer.

Part of the observed difference in baseline LE differences may be attributable to the fact that most of the CTs in the treated group are in eastern suburban neighbourhoods, where there may be less household deprivation but also less access to public services, such as health care and transportation, compared to the control group where CTs were found throughout the east and west of the study area. This limitation of the ‘Ashenfelter Dip’ [62,63], whereby participants may be systematically different than non-participants, is common among DID designs. The advantage of using the DID design is that participants may start at different levels of the outcome (health) and covariates (services); the focus of the DID is to assess the changes in the outcome, with the assumptions that time-varying confounders are group invariant and group-varying confounders are time invariant. Follow up studies can be conducted to assess the density-deprivation relationship and proximity to essential services to further tease out the differential findings by SES. Furthermore, future studies can also consider the use of a ’difference in difference in difference (DDD) design’, which is a multi-group, multi-time period design whereby an additional comparison group is added to address any potential group or time-varying confounders of concern, such as changes in SES over time [64].

Overall, our study attempts to quantitatively evaluate the optimal ranges with a methodology that can be easily replicated in other cities for comparison, and which can also incorporate important sociodemographic information. Our use of age standardized mortality rates is a metric that is widely used and more interpretable for the wider public. Future studies should also consider reporting relative risk and disaggregate further between different vulnerable subgroups, ethnicities, and age groups. Considerations of these populations for future urban land use and design is especially important given established public health inequities among ethnic populations [65,66] and the growing senior populations [67]. Putting in the forethought of universal neighbourhood designs and targeted interventions by vulnerable subgroups can prevent the exacerbation of these inequalities.

## 5. Conclusions

In summary, this study presents novel findings from using a recently developed 27-year intra-urban measures to model the shape of the relationship of multiple density-mortality functions and to tease out the effects of densification over time. These findings reveal that areas which are often associated with higher population-densities, such as the downtown core, provide health benefits from the major causes of mortality, but these environments also associated with higher relative inequality from increases in largely preventable deaths. Planners should take these mixed findings by cause into consideration and find solutions to offset the health penalties from increasing densification. For example, if there are higher inequalities in areas of high density but are generally more protective from major causes of mortality, planners can consider the strategic implementation of new and/or improvement of existing programs that are associated with these causes. These can include methadone clinics, HIV/STI testing centers, maternal–child health programs, cancer screening, and/or experiments with automotive regulations, such as the banning of private cars or ‘congestion charges’ in city centers that have been proposed in major cities, such as London, Madrid, and Venice. Finding the ‘optimal balance’ of densification and services will be especially important for initiatives such as the ’15 min city’, where higher densities have been promoted to improve access and proximity of urban amenities.

Overall, densification has been contentious, especially during infectious disease epidemics and with chronic housing shortages and affordability concerns in major metropolitan cities around the world. Is the recent pandemic a turning point for urban planning as it relates to densification [52]? We hope that the findings in this study continue the important narrative that careful densification is needed and should be prioritized on the sustainability agenda as urban population growth accelerates. The narrative should include the important social, economic, and health benefits and costs trade-offs that are made with increased densification and the interventions required to offset the urban health penalties. This study is only the first step in Vancouver and one of multiple initiatives needed to produce more nuanced understanding of the thresholds of densification in various contexts. Importantly, future studies should consider how inequality can be exacerbated from densification for the most vulnerable neighbourhoods and subgroups, especially as the population of Metro Vancouver is projected to reach over 3.4 million by 2041 [28]. If densification is planned carefully, and we continue to refine our understanding of the trade-offs and optimal ranges of populations, employments, and services, we can transition and develop cities to be prosperous, sustainable, and healthy.

## Figures and Tables

**Figure 1 ijerph-19-02900-f001:**
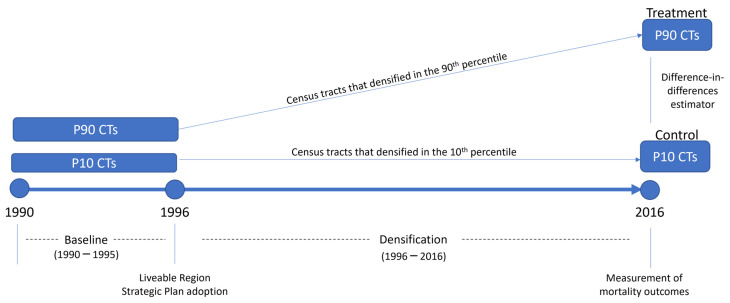
Study design of treatment (P90 CTs) and control (P10 CTs) groups.

**Figure 2 ijerph-19-02900-f002:**
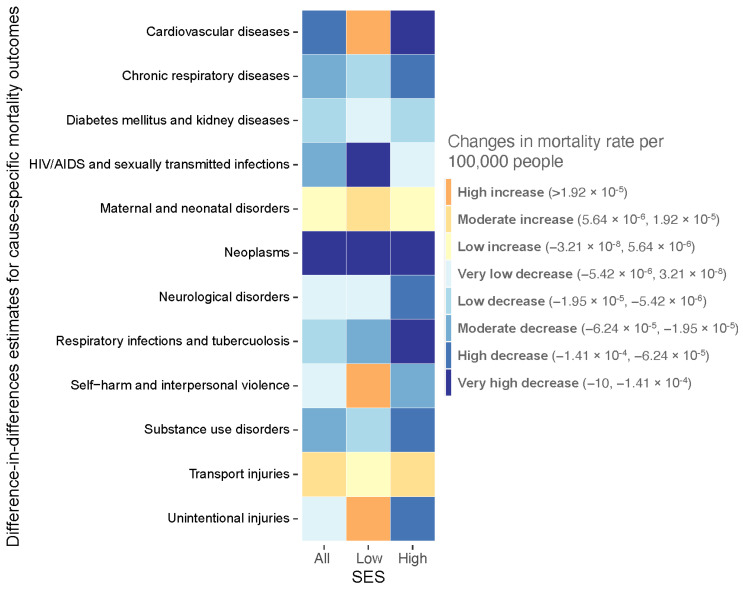
Summary of the direction and relative magnitude of difference-in-differences estimates on cause-specific mortality rates by SES (All CTs, Low SES CTs, and High SES CTs), Metro Vancouver, British Columbia, 1990–2016. Difference-in-differences estimates were categorised into tertiles for increases [high increase (>1.92 × 10^−5^ per 100,000 people), moderate increase (low to high range of tertile: 5.64 × 10^−6^, 1.92 × 10^−5^), and low increase (−3.21 × 10^−8^, 5.64 × 10^−6^)] and quintiles for decreases [very-low decrease (low to high range of quintile: −5.42 × 10^−6^, 3.21 × 10^−8^), low decrease (−1.95 × 10^−5^, −5.42 × 10^−6^), moderate decrease (−6.24 × 10^−5^, −1.95 × 10^−5^), high decrease (−1.41 × 10^−4^, −6.24 × 10^−5^), and very-high decrease (−10, −1.41 × 10^−4^)].

**Figure 3 ijerph-19-02900-f003:**
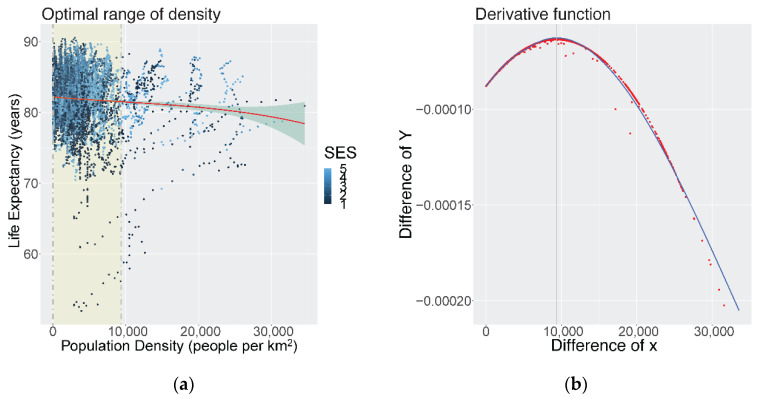
(**a**) Smoothed cubic spline with the shaded area representing the ‘optimal range’ identified of population density (people per km^2^) and life expectancy at birth, Metro Vancouver, 1990–2016; (**b**) The first order derivative to identify the inflection point and threshold of densification.

**Figure 4 ijerph-19-02900-f004:**
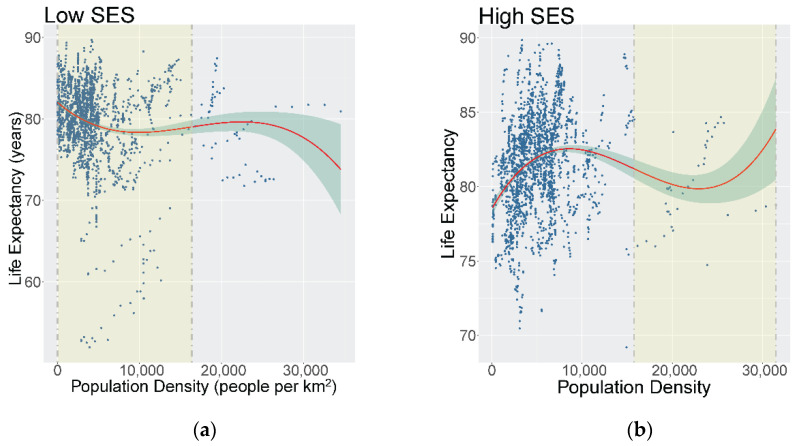
(**a**) Smoothed cubic spline population density (people per km^2^) and life expectancy at birth for low SES CTs with the shaded areas representing the ‘optimal ranges’ identified, Metro Vancouver, 1990–2016; (**b**) Smoothed cubic spline population density (people per km^2^) and life expectancy at birth for high SES CTs with the shaded areas representing the ‘optimal ranges’ identified, Metro Vancouver, 1990–2016.

**Figure 5 ijerph-19-02900-f005:**
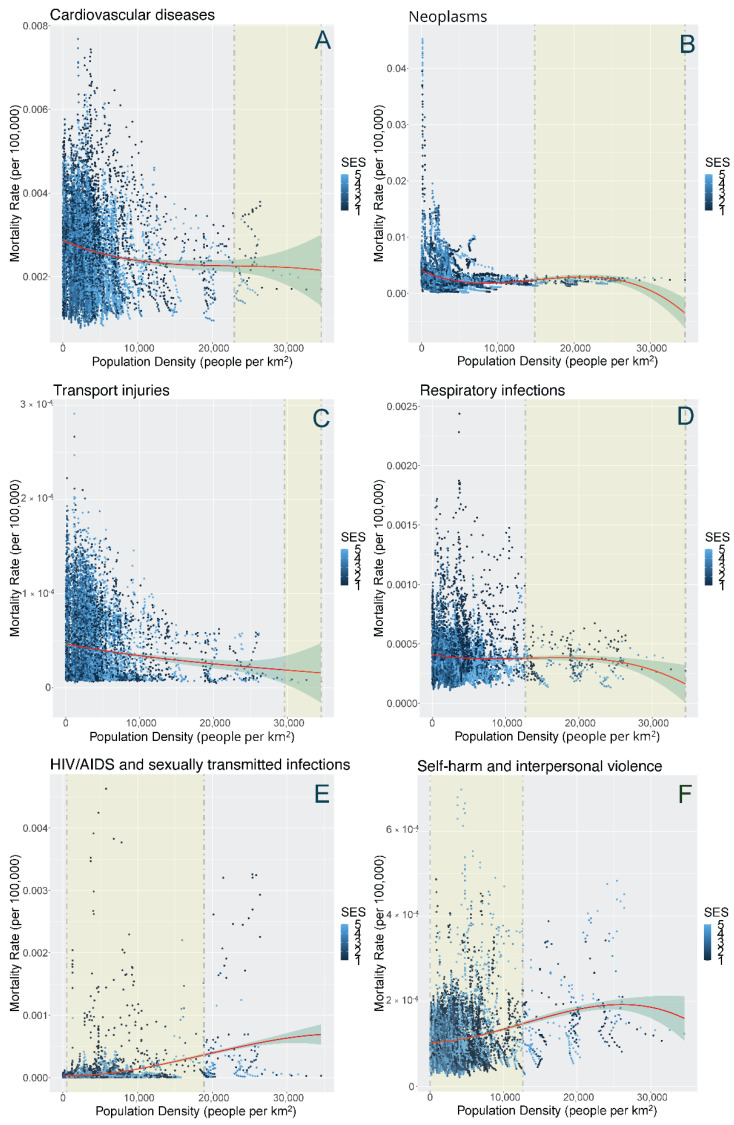
Smoothed cubic splines with the shaded areas representing the ‘optimal range’ identified of population density (people per km^2^) and six cause-specific mortality rates: (**A**) cardiovascular diseases; (**B**) neoplasms; (**C**) transport injuries; (**D**) respiratory infections; (**E**) HIV/AIDS and sexually transmitted infections; and (**F**) self-harm and interpersonal violence, Metro Vancouver, 1990–2016.

**Table 1 ijerph-19-02900-t001:** Baseline descriptive characteristics of census tracts (CTs) in the entire study region and in each of the three difference-in-differences models (all CTs, low socioeconomic status (SES) CTs, and high SES CTs), Metro Vancouver, 1990–1995 ^a^.

	No. of CTs	Median Area (km^2^)	Median Population Density (People per km^2^)	Median Life Expectancy (Years)	Median Material Deprivation Index Score	Median Social Deprivation Index Score
**Entire study region**	368	1.8	2463	79.8	1.03	1.02
**Model 1: All CTs** ^b^	74	2.6	713	79.7	1.05	1.03
P90 CTs	37	3.1	387	79.6	1.05	1.05
P10 CTs	37	2.2	1915	79.8	1.05	1.02
**Model 2: Low SES (high MSDI) CT subset** ^b^	39	1.3	715	78.6	625.90	625.95
P90 CTs	19	1.4	515	79.0	763.67	763.66
P10 CTs	20	1.3	2974	78.4	331.27	331.19
**Model 3: High SES (low MSDI) CT subset** ^b^	33	2.2	2793	77.2	0.99	0.98
P90 CTs	17	3.2	1482	75.7	1.00	0.95
P10 CTs	16	1.6	5802	78.3	0.99	0.99

^a^ All variables (except area) are point-based summary measures for pre-LRSP periods. ^b^ Included are CTs that met the inclusion criteria: 90th percentile and 10th percentile densification changes within (1) all CTs; (2) low SES CTs (lowest quintile group of MSDI scores); and (3) high SES CTs (highest quintile group of MSDI scores).

## Data Availability

Census data can be downloaded from Statistics Canada. Mortality data can be found at the University of British Columbia Environmental Health Research Team website. Deprivation data can be retrieved from Canadian Urban Environmental Health Research Consortium.

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
