# Peer review of "Assessing Trade-Offs and Optimal Ranges of Density for Life Expectancy and 12 Causes of Mortality in Metro Vancouver, Canada, 1990–2016"

_ijerph, 2022, doi:10.3390/ijerph19052900_

Round 1

Reviewer 1 Report

The manuscript brings a very important discussion regarding human density in urban areas in relation to health and to growth management plans in Greater Vancouver. It is very up to date regarding a strong ecological movement in favour of compact cities and the challenges regarding crowded cities and the possible vulnerability to infectious diseases and to transit accidents. The paper is based on qualified data and uses models which take in account SES - stratification by economic status. The approach of finding DID - difference in differences is innovative. The results are interesting and important for city planning, in spite of being mostly descriptive and little explorative of the causes. There are other factors that might have influenced results, besides human density, and were not included in the model or in the discussion. As examples, I mention air pollution, police action, density of health services.

In table 1, Model 2, why median deprivation scores are so different than in models 1 and 3?

Study limitations are well described.

Literature used in the discussion is very up to date and of good quality.

Author Response

Response to Reviewer 1 Comments

Thank you for your detailed comments and feedback to help improve this paper. We have responded to the specific comments and revised the manuscript accordingly as discussed below.

Point 1: The manuscript brings a very important discussion regarding human density in urban areas in relation to health and to growth management plans in Greater Vancouver. It is very up to date regarding a strong ecological movement in favour of compact cities and the challenges regarding crowded cities and the possible vulnerability to infectious diseases and to transit accidents. The paper is based on qualified data and uses models which take in account SES - stratification by economic status. The approach of finding DID - difference in differences is innovative. The results are interesting and important for city planning, in spite of being mostly descriptive and little explorative of the causes. There are other factors that might have influenced results, besides human density, and were not included in the model or in the discussion. As examples, I mention air pollution, police action, density of health services.

Response 1: We agree that there are many other factors that might influence the results we observed. We believe this study initiates the first step and we hope that future studies will assess additional explanatory factors including the ones you listed. We added some reflection on this topic in the Discussion section:

“Over time, density decreased mortality from these causes in high SES CTs but increased mortality in low SES CTs. In fact, density decreased mortality rates for almost all causes in higher magnitudes in high SES CTs compared to low SES CTs, except for HIV/AIDS and STIs and transport injuries; the latter cause group increased for all CTs regardless of SES status, which is unsurprising given that there are increased opportunities to interact with vehicles with more people.”

“These differences may be attributed to the inequitable types of services, employment, and access to natural space that is available with increasing density by SES. For example, the CVD mortality outcomes observed in high SES CTs may be due to well-reported linkages to health-promoting mediators that are available in these CTs, such as increased opportunities for leisure and work-related physical activity [22,34], transit access [35], and nutritional food options [36,37].  In contrast, high dense and low SES CTs may also be linked to higher densities of fast food outlets [38,39], injury related to violence [40], and crowding [41]. Other spatially varying factors such as air pollution which is typically correlated with density may also be relevant [42].”

“Previous studies have indicated that mortality from chronic respiratory diseases is related with higher population density and smoking status but varies by age [50]; higher rates of chronic obstructive pulmonary deaths are more often found among younger populations in urban areas and senior populations in rural areas. Future studies should examine these functions with age-specific rates to further unravel these relationships.”

As well, in the limitations paragraph:

“Additional research on population density and the inter-dependencies with other spatial attributes like densities of various housing types, employment, services, nighttime population, or occupancy, would provide more nuanced understanding of the overall density-health relationship [6].”

Point 2: In table 1, Model 2, why median deprivation scores are so different than in models 1 and 3?

Response 2: We applied a differences-in-differences model for all CTs in model 1 (no SES stratification was done). In model 3, we applied a DID model for only high SES CTs. In model 2, we applied a DID model for only low SES CTs, which explains why the deprivation scores are much higher than in model 1 and 3. We reported the non-standardised numeric factor scores in the descriptive tables, which explains the large range in the scores (1-700+). For stratification and modelling purposes, we standardised and categorised the scores into quintiles.

Study limitations are well described.

Literature used in the discussion is very up to date and of good quality.

Reviewer 2 Report

I have no suggestions for authors. In my opinion, this is a very interesting and well-written article.

Author Response

Response to Reviewer 2 Comments

Point 1: I have no suggestions for authors. In my opinion, this is a very interesting and well-written article.

Response 1: Thank you for your positive feedback. We are glad that you found the article interesting.

Reviewer 3 Report

Interesting paper. Minor comments below:

Changes in Population density, life expectancy and mortality may be affected by people movements during the `deification’ period. For example: 1) increase in international immigrants (selected population with advantages in health and socioeconomical status, 2) population living in P10CTs in the 1990s may move to P90CTs during the ‘Densification’ period or the other way around due to individual circumstances.

Changes in Population density, life expectancy and mortality may be affected by changes in ‘SES’ in addition to ‘SES’ at baseline.

The associations between density changes and live expectancy, mortality may not be linear. For example, while the baseline level of live expectancy and mortality rates varied in different suburbs, the potential limited improvement in live expectancy and mortalities is largely determined by the advancement of healthcare and science. For example, improvement of live expectancy from 70 to 75 is likely to be much easier than increasing live expectancy from 85 to 90.

The variations of the estimated mortality rates as well as changes in the rates are greatly affective by the size of population at risk especially for disease with low incidence rates, which is appeared to be the case as indicated by figure 5. Difference in mortality rates may be modelled using ratios (relative risk), and log transform of the rates may help to reduce the variations.

It would be helpful if the authors could provide a list of variables controlled in the models.

Author Response

Response to Reviewer 3 Comments

Interesting paper. Minor comments below:

Thank you for your detailed comments and feedback to help improve this paper. We have responded to the specific comments and revised the manuscript accordingly as discussed below.

Point 1: Changes in Population density, life expectancy and mortality may be affected by people movements during the `deification’ period. For example: 1) increase in international immigrants (selected population with advantages in health and socioeconomical status, 2) population living in P10CTs in the 1990s may move to P90CTs during the ‘Densification’ period or the other way around due to individual circumstances.

Response 1: We agree that population mobility can impact CT cumulative exposure and therefore lead to potential misclassification, especially of people’s earlier experiences. We have decided to include some discussion on this in the limitations section:

“Population mobility during the densification period can impact CT cumulative exposure and therefore lead to potential misclassification, especially of people’s earlier experiences. This potential bias is especially relevant for the younger populations more than older populations who are more settled. For example, younger migrants have been shown to move longer distances; the migration out of the CT has an effect of decreasing the proportion of healthier populations in the area. However, older migrants tend to move shorter distances to similar environments where there are good medical services. In a previous migration analysis of this mortality dataset [30], we linked 10 years of residential history to the mortality data and assessed how reassigning the CT of exposure based on duration of residency compared to last recorded CT changed the life expectancy measures. The analysis did not show a substantial change in the LE estimates. Nevertheless, mobility of inter-national migrants, who tend to be healthier, and populations moving between CTs of different SES were not accounted for. Unfortunately, annual residential data beyond this temporal scope was not available. Future collection and ongoing surveillance of self-reported residential histories especially during earlier years can potentially help address this limitation.

Point 2: Changes in Population density, life expectancy and mortality may be affected by changes in ‘SES’ in addition to ‘SES’ at baseline.

Response 2: We agree with this observation. One of the advantages of using the DID design (unlike a randomized controlled trial) is that CTs may start at different levels of the outcome (health) and covariates (SES). The focus of the DID is to assess the changes in the outcome (health) from changes in the environmental conditions (e.g., population densification), with the assumptions that time-varying confounders (such as age) are group invariant and group-varying confounders (such as SES) are time invariant. In essence, this design aims to account for unmeasured confounders. Nevertheless, we agree that it is an important factor and stratified our analysis by baseline SES to observe the effect of densification in each SES group.

Overall, the goal of this study was to initiate the first step in assessing the effects of densification on multiple health outcomes – a two group, two time period design suited our needs. Future studies can build upon the initial results of this study to further unravel and explain our findings. This can include using a multi-group, multi-time period ‘differences in difference in differences (DDD)’ design, which can add additional comparison groups to address any confounders of concern, such as SES.

We have added some reflections about this in the Limitations section:

“This limitation of the ‘Ashenfelter Dip’ [62,63], whereby participants may be systematically different than non-participants, is common among DID designs. The advantage of using the DID design is that participants may start at different levels of the outcome (health) and covariates (services); the focus of the DID is to assess the changes in the outcome, with the assumptions that time-varying confounders are group invariant and group-varying confounders are time invariant. Follow up studies can be conducted to assess the density-deprivation relationship and proximity to essential services to further tease out the differential findings by SES. Furthermore, future studies can also consider the use of a ’difference in difference in difference (DDD) design’, which is a multi-group, multi-time period design whereby an additional comparison group is added to address any potential group or time-varying confounders of concern, such as changes in SES over time [64].”

Point 3: The associations between density changes and live expectancy, mortality may not be linear. For example, while the baseline level of live expectancy and mortality rates varied in different suburbs, the potential limited improvement in live expectancy and mortalities is largely determined by the advancement of healthcare and science. For example, improvement of live expectancy from 70 to 75 is likely to be much easier than increasing live expectancy from 85 to 90.

Response 3: Thank you for that observation and we agree that the density to mortality/life expectancy relationship may not be linear. With this assumption, we would expect a CT with a lower baseline median life expectancy to have a greater improvement in life expectancy from densification when compared with a CT with higher median life expectancy. Although the baseline life expectancy for the first two models were approximately similar, there were some differences for the third model (~2.6 years lower LE for P90CTs compared to P10CTs). We have added some text in the Limitations section to reflect this discussion:

“Moreover, for the DID models, there were baseline LE differences for the high SES CTs of around 2.7 years lower in the treatment group compared to the control group, meaning the treatment group started at a lower life expectancy before densification. A large difference in baseline LE may affect the DID results if we expect that improvements in life expectancy from densification is not linear. That is, public and environmental health interventions may be more likely to improve life expectancy from 70 to 75 years than from 85 to 90 years, as an example. For latter age groups, we would expect greater contribution and advancements from healthcare and science to help people live longer.”

Point 4: The variations of the estimated mortality rates as well as changes in the rates are greatly affective by the size of population at risk especially for disease with low incidence rates, which is appeared to be the case as indicated by figure 5. Difference in mortality rates may be modelled using ratios (relative risk), and log transform of the rates may help to reduce the variations.

Response 4: Although we had considered log transforming the rates and reporting relative risk, we ultimately decided to report the age standardised mortality rates given it’s a widely used metric by healthcare and urban planners and easier to interpret for the wider general public. However, we added a line in the limitations for consideration in future studies:

“Overall, our study attempts to quantitatively evaluate the optimal ranges with a methodology that can be easily replicated in other cities for comparison, and which can also incorporate important sociodemographic information. Our use of age standardized mortality rates is a metric that is widely used and more interpretable for the wider public. Future studies should also consider reporting relative risk and disaggregate further between different vulnerable subgroups, ethnicities, and age groups. Considerations of these populations for future urban land use and design is especially important given established public health inequities among ethnic populations [65,66] and the growing senior populations [67]. Putting in the forethought of universal neighbourhood designs and targeted interventions by vulnerable subgroups can prevent the exacerbation of these inequalities.”

Point 5: It would be helpful if the authors could provide a list of variables controlled in the models.

Response 5: We used a simple two group, two time period DID design stratified by SES for this study given the lack of spatial data that was available at this resolution (CT) and temporal period (27 years). Therefore, no other variables were adjusted for in these models.

Reviewer 4 Report

The authors provide a very relevant and detailed analysis of the impact of population density on life expectancy and mortality. It is well written and minor corrections are listed below:

Line 141: How were the six indicators chose to identify social deprivation?

Line 145: Was land use change accounted for in addition to population density change? It seems that some areas may densify by simply adding more of the same type dwelling units (i.e., more apartment buildings), while other areas may densify by changing dwelling units (i.e., from single-family to apartment buildings).

Line 201: Consider "half" as opposed to "2 times lower".

Line 204: Consider "one third" (or "nearly a quarter") as opposed to "3.9 times lower".

Table 1: The column header font is not consistent. Same issue for the first column.

Figure 2 is difficult to interpret. What is the y-axis?

Figures 4 a. and b. would benefit from similar axes.

Line 298: "2" should be superscripted in "km2" (check the entire document for this).

Figure 5: The caption is not in order - it would be better if each panel was labeled (i.e., a., b., c., etc.).

There doesn't have to be blank lines between paragraphs.

Line 412: Should "old" be "older"? The sentence could be rewritten for clarity.

Line 436 (and elsewhere): The correct spelling is: "COVID-19".

Line 479: It would be helpful to remind the reader whether the 2.7 years was positive or negative.

Section 4 should be broken up into subsections to separate the Discussion section. 4.1 is not necessary if there is no 4.2, etc.

Line 494: "Evaluated" should be "evaluate".

Line 499: Is "publish" supposed to be "public"?

Line 511: "are" is missing from the sentence.

Line 519: "in" is missing before "major cities".

Author Response

Response to Reviewer 4 Comments:

The authors provide a very relevant and detailed analysis of the impact of population density on life expectancy and mortality. It is well written and minor corrections are listed below:

Thank you for your detailed comments and feedback to help improve this paper. We have responded to the specific comments and revised the manuscript accordingly as discussed below.

Point 1: Line 141: How were the six indicators chose to identify social deprivation?

Response 1: More information about the index can be read in Pampalon et al.’s 2012 paper. We did not derive the social deprivation index ourselves but rather applied it in our analysis. It has been widely used in epidemiologic analyses in Canada. Data were drawn from the Canadian census and combined well with our analysis at the census tract level. Briefly, the Material and Social Deprivation Index combines six different variables chosen to reflect material deprivation, or the lack of everyday goods and commodities, and social deprivation, the fragility of an individual’s social network. The definitions are derived from the ideas of Peter Townsend, who conceptualised both material and social deprivation as, “a state of observable and demonstrable disadvantage relative to local community or the wider society or nation to which the individual, family or group belongs”. The three variables for the material deprivation component include: persons without high school diploma, ratio employment/population, and average personal income. The three variables for the social deprivation component include: persons living alone, persons separated, divorced or widowed, and single-parent families. More details were added to this paragraph in the Methods section:

“The Canadian Material and Social Deprivation Index (MSDI) was based on the Canadian census and was collected from the Canadian Urban Environmental Health Research Consortium (CANUE) [32]. Six different variables were chosen based on Peter Townsend’s idea of ‘observable and demonstrable disadvantage’ that reflect material deprivation, or the lack of everyday goods and commodities, and social deprivation, the fragility of an individual’s social network. The material deprivation variables include persons without high school diploma, ratio employment/population, and average personal income. The social deprivation variables include persons living alone, persons separated, divorced or widowed, and single-parent families.”

Point 2: Line 145: Was land use change accounted for in addition to population density change? It seems that some areas may densify by simply adding more of the same type dwelling units (i.e., more apartment buildings), while other areas may densify by changing dwelling units (i.e., from single-family to apartment buildings).

Response 2: No, land use was not accounted for in this assessment, but we agree it is an important urban planning metric that would be helpful in this discussion and provide more nuanced understanding of density.  We added more discussion in our existing paragraph about this topic in our Discussion paragraph:

“Population density is only one type of discrete, two-dimensional areal measure for planners to use [58]. It includes only daytime population of residential neighbourhoods, thereby excluding daily changes in density levels based on employment and educational centers. Moreover, density changes in this study did not differentiate between land use changes. That is, did neighbourhoods increase population through the addition of the same dwelling units and/or from changes in land permits, such as from single-family zoning to multi-family zoning? Future studies should consider more nuanced under-standing of density, including incorporating measures such as measured or perceived crowding, ‘buffer zones’ around the census output area, land use changes, or floor area space [59,60]. Future studies can also consider net density instead of gross density by including areas that are only residential or mixed-use, thereby excluding commercial areas and parks. Although, the measurement of net density varies across different settings, which makes it difficult for universal application [7]. Arguably, gross density can still be considered a useful summary measure as it is often correlated with several well-known built environmental factors, such as service availability, mixed land use, and lack of natural space [7,61]. Additional research on population density and the inter-dependencies with other spatial attributes like densities of various housing types, employment, services, nighttime population, or occupancy, would provide more nuanced understanding of the overall density-health relationship [6].”

Point 3: Line 201: Consider "half" as opposed to "2 times lower".

Response 3: This change was implemented as suggested.

Point 4: Line 204: Consider "one third" (or "nearly a quarter") as opposed to "3.9 times lower".

Response 4: This change was implemented as suggested.

Point 5: Table 1: The column header font is not consistent. Same issue for the first column.

Response 5: The font has been changed to ensure consistency.

Point 6: Figure 2 is difficult to interpret. What is the y-axis?

Response 6: The y-axis has been labeled (Difference-in-differences estimates for cause-specific mortality outcomes).

Point 7: Figures 4 a. and b. would benefit from similar axes.

Response 7: The axes of Figure 4B has been changed to have a similar axis to Figure 4A.

Point 8: Line 298: "2" should be superscripted in "km2" (check the entire document for this).

Response 8: This change was implemented as suggested.

Point 9: Figure 5: The caption is not in order - it would be better if each panel was labeled (i.e., a., b., c., etc.).

Response 9: The caption has been changed and each panel was labeled as suggested.

Point 10: There doesn't have to be blank lines between paragraphs.

Response 10: Blank lines were removed between paragraphs.

Point 11: Line 412: Should "old" be "older"? The sentence could be rewritten for clarity.

Response 11: This line was rewritten for clarity:

“Previous studies have indicated that mortality from chronic respiratory diseases is related with higher population density but varies by age [50]; higher rates of chronic obstructive pulmonary deaths are more often found among younger populations in urban areas and senior populations in rural areas.”

Point 12: Line 436 (and elsewhere): The correct spelling is: "COVID-19".

Response 12: This change was implemented as suggested.

Point 13: Line 479: It would be helpful to remind the reader whether the 2.7 years was positive or negative.

Response 13: This line was rewritten for clarity:

“Moreover, for the DID models, there were baseline LE differences for the high SES CTs of around 2.7 years lower in the treatment group compared to the control group, meaning the treatment group started at a lower life expectancy before densification.”

Point 14: Section 4 should be broken up into subsections to separate the Discussion section. 4.1 is not necessary if there is no 4.2, etc.

Response 14: Three subheadings have been added in the Discussion section as suggested.

Point 15: Line 494: "Evaluated" should be "evaluate".

Response 15: This change was implemented as suggested.

Point 16: Line 499: Is "publish" supposed to be "public"?

Response 16: This change was implemented as suggested.

Point 17: Line 511: "are" is missing from the sentence.

Response 17: This change was implemented as suggested.

Point 18: Line 519: "in" is missing before "major cities".

Response 18: This change was implemented as suggested.
